# The Selective CB2 Agonist COR167 Reduced Symptoms in a Mice Model of Trauma-Induced Peripheral Neuropathy through HDAC-1 Inhibition

**DOI:** 10.3390/biomedicines11061546

**Published:** 2023-05-26

**Authors:** Vittoria Borgonetti, Claudia Mugnaini, Federico Corelli, Nicoletta Galeotti

**Affiliations:** 1Department of Neuroscience, Psychology, Drug Research and Child Health (NEUROFARBA), Section of Pharmacology and Toxicology, University of Florence, Viale G. Pieraccini 6, 50139 Florence, Italy; vittoria.borgonetti@unifi.it; 2Department of Biotechnology, Chemistry and Pharmacy, University of Siena, Via Aldo Moro 2, 53100 Siena, Italy; claudia.mugnaini@unisi.it (C.M.); federico.corelli@unisi.it (F.C.)

**Keywords:** CB2 receptor agonist, cannabinoid system, neuropathic pain, microglia, HDAC1

## Abstract

Neuropathic pain is a chronic disabling condition with a 7–10% of prevalence in the general population that is largely undertreated. Available analgesic therapies are poorly effective and are often accompanied by numerous side effects. Growing evidence indicates cannabinoids are a valuable treatment opportunity for neuropathic pain. The endocannabinoid system is an important regulator of pain perception through the CB1 receptors, but CB1 agonists, while largely effective, are not always satisfactory pain-relieving agents in clinics because of their serious adverse effects. Recently, several CB2 agonists have shown analgesic, anti-hyperalgesic, and anti-allodynic activity in the absence of CB1-induced psychostimulant effects, offering promise in neuropathic pain management. The aim of this study was to evaluate the anti-neuropathic activity of a novel selective CB2 agonist, COR167, in a preclinical model of peripheral neuropathy, the spared nerve injury (SNI). Oral COR167, in a dose-dependent manner, attenuated mechanical allodynia and thermal hyperalgesia after acute and repeated administration, showing the absence of tolerance induction. At anti-neuropathic doses, COR167 did not show any alteration in the locomotor behavior. SNI mice showed increased microglial levels of HDAC1 protein in the ipsilateral side of the spinal cord, along with NF-kB activation. COR167 treatment prevented the HDAC1 overexpression and the NF-kB activation and increased the levels of the anti-inflammatory cytokine IL-10 through a CB2-mediated mechanism. Oral administration of COR167 shows promising therapeutic potential in the management of neuropathic pain conditions.

## 1. Introduction

Neuropathic pain is a chronic worldwide disease due to an injury to the somatosensory system, which drastically worsens the patient’s quality of life [1]. So far, there are few effective therapies, which are generally characterized by numerous side effects. In seeking more effective innovative therapies, one should take into account the endocannabinoid system, a key regulator of chronic pain, especially through modulation of its main receptors, CB1 and CB2 [2,3]. Even though CB1 stimulation induced analgesia in several painful states [4,5,6,7], CB1 agonists are not ideal pain-relieving agents for clinical use due to their serious side effects, such as addiction, excessive sedation, fatigue, and dizziness [8,9]. Thus, the alternative of using CB2 receptor modulators could be a therapeutic advantage, as they lack the negative neurological effects induced by CB1 receptor modulation. The identification of CB2 receptors in glial cells has opened new therapeutic approaches for these ligands in chronic pain management [10,11,12]. Indeed, CB2 are predominantly present in peripheral and central immune cells and, even if their presence in neurons is still controversial, by modulating the immune system, they exert anti-inflammatory effects [13,14,15] and, thus, a neuroprotective activity [16,17,18,19,20]. Findings reported that lesions to peripheral nerves increased the expression of CB2 in the dorsal root ganglia [21] and the spinal cord tissue [13,22] of rodents with neuropathy and CB2 agonists exerted anti-hyperalgesic effects [21,23,24,25,26] through the reduction of neuroinflammation and microgliosis. Selective CB2 agonists could be potentially used to relieve pain, thus circumventing the psychostimulant side effects of CB1 agonists.

The main aim of this work is to investigate the possible anti-hyperalgesic activity of a novel CB2 selective agonist, COR167, in an animal model of peripheral neuropathy, the spared nerve injury (SNI). Previously, COR167 showed neuroprotective activity [27], counteracted glial tumor growth [28], exerted an analgesic effect in the formalin test of acute peripheral and inflammatory pain [29], and had anti-neuroinflammatory activity [30]. It has been reported that damage to peripheral nerves induces an increase in the expression of CB2 receptors in primary sensory neurons through histone modifications [21]. Moreover, non-psychotropic *Cannabis sativa* L. oil attenuated peripheral neuropathy symptoms in a mouse model through the modulation of CB2 and the reduction of neuropathy-induced HDAC1 overexpression [13]. Histone deacetylase 1 (HDAC1) is an enzyme involved in pathophysiological processes related to microglial activation in neuropathic pain by modulating inflammatory processes. In fact, an increase in its expression has been observed in spinal cord samples of animals with neuropathy [31,32], and HDAC inhibitors are widely reported in the literature for reducing chronic pain [33]. Here, we investigated whether COR167 activity may be related to the modulation of HDAC1 both in the spinal cord of mice with neuropathy and in an in vitro model of neuroinflammation in microglial BV2 cells.

## 2. Materials and Methods

### 2.1. Drugs Administration

COR167 (N-(adamantan-1-yl)-6-isopropyl-4-oxo-1-pentyl-1,4-dihydroquinoline-3-carboxamide), a selective CB2 agonist, was synthesized as previously reported [29]. COR167 was dissolved in 5% DMSO and orally administered at increasing concentrations (3, 10, 30, and 100 mg/kg). Pregabalin (PREG 30 mg/kg i.p.; Sigma Aldrich, Milan, Italy), used as a reference drug for mechanical allodynia, was dissolved in saline solution and administered 3 h before testing. Morphine (MORPH 7 mg/kg i.p.; SALARS, Como, Italy), used as a reference drug in the thermal hyperalgesia assay, was dissolved in saline. SAHA (suberoylanilide hydroxamic acid) (10 mg/kg i.p.; Sigma-Aldrich, Milan, Italy), used as a reference HDACs inhibitor [34], was dissolved in 5% DMSO.

### 2.2. Animals

CD1 male mice (20–22 g; Envigo, Varese, Italy) were used. Animals were kept at 4–5 animals per cage under controlled environmental conditions (23 ± 1 °C, 12 h light/dark cycle, lights on from 7:00 a.m. to 7:00 p.m., and access to water and food ad libitum). All experimental procedures and animal care complied with international laws and policies (Directive 2010/63/EU of the European Parliament) under license from the Italian Department of Health (54/2014-B, 410/2017-PR). Behavioral studies complied with animal research reporting of in vivo experiments (ARRIVE) guidelines [35,36]. All efforts were made to minimize the number of animals used and their suffering. Mice were sacrificed by cervical dislocation for removal of the spinal cord to perform in vitro analysis. The number of mice employed per experiment was identified on power analysis (G power software) [37].

### 2.3. Spared Nerve Injury (SNI) Model

The SNI surgical procedure was performed as described [38]. Mice were anesthetized with a mixture of 4% isoflurane in O_2_/N_2_O (30:70 *v*/*v*) and placed in a prone position. 

Then, a 5 mm incision was made on the right thigh limb (commonly named *ipsi*) to expose the sciatic nerve and its three branches. The tibial and common peroneal nerves were ligated and transected together, while the sural nerve was preserved. For the sham control group, mice underwent the same manipulation except for nerve ligation and nerve transection. The left paw was left unaltered (commonly named *contra*). The average absolute threshold (g) was calculated by subtracting contra and ipsi values registered during the time course. We used two different cohorts of animals: the first (Cohort 1 *n* = 40) was used for the dose-response curve, while the second was made up of only a group for the repeated oral administration of the best active dose (Cohort 2 *n* = 24).

### 2.4. Pain Hypersensitivity

#### 2.4.1. Von Frey Filaments

The von Frey test was used to evaluate mechanical allodynia [39]. The tests were carried out before the operations (reference baseline values) and afterward. Sensitivity to a mechanical stimulus was measured by von Frey monofilaments. Mice were placed individually in plexiglass test cages [8.5 × 3.4 × 3.4 (h) cm] with metal mesh floors and were allowed to acclimate to their surrounding for 1 h before testing. The von Frey monofilaments with increasing degrees of strength (0.04, 0.07, 0.16, 0.4, 0.6, 1.0, 1.4, 2.0 g) were applied to the skin on the lateral side of the paw sole on both ipsilateral and contralateral sides. Any nocifensive behavior exhibited by the mouse was considered a positive response. If a negative response occurred, the adjacent larger next filament was used, and testing continued until three over five positive responses were collected after the first response change. The averages of the responses were finally calculated.

#### 2.4.2. Hot Plate Test

The hot plate test was used to evaluate the thermal hyperalgesia through a hot plate analgesiometer maintained at 52.5 ± 0.1 °C and performed as described [40]. The latency time (s) to the response of the animals to the thermal stimulus (shaking or licking their hind paw) was measured. An arbitrary cut-off time of 45 s was adopted.

### 2.5. Locomotor Activity

#### 2.5.1. Rotarod Test

Possible side effects of COR167 on motor performance were assessed by the rotarod test [41]. The number of falls in 30 s was counted and used as an indication of locomotor coordination. 

#### 2.5.2. Hole-Board Test

The hole-board test was used to evaluate spontaneous locomotor activity [41]. Each mouse was tested individually over a period of 5 min. The spontaneous mobility was determined by registering the movements of each animal on the plane by means of 4 photo beams crossing the plane from midpoint to midpoint of opposite sides. The exploratory activity of mice was evaluated with miniature photoelectric cells contained in each hole, registering the head–hips of each mouse.

### 2.6. Evaluation of the Anxiolytic-like Effect

#### Open Field Test

This test evaluated the animals’ anxiety-like behavior [38]. Briefly, animals were positioned in the center of a rectangular box (78 × 60 × 39 cm), and then the time it remained in the internal portion was measured, compared to a total duration of 5 min. A longer permanence of the animal in the center of the arena was taken as an indication of low levels of anxiety. This test was performed in the baseline condition (before surgery) and on post-surgical days 7, 14, and 21.

### 2.7. Tissue Protein Extraction

In order to detect protein expression in the animals’ tissues, spinal cords were removed on day 14 post-surgery by separating the contralateral and ipsilateral sides. Samples were homogenized in a lysis buffer containing 25 mM Tris-HCl pH (7.5), 2.5 mM EDTA, 5 mM EGTA, 25 mM NaCl, 4 mM PNFF, 2 mM NaPP, 1 mM PMSF, 1 mM di Na_3_VO_4_, 50 μg/mL aprotinin, 20 μg/mL leupeptin, and 0.1% SDS (Merck, Darmstadt, Germany). The homogenate was centrifuged at 12,000× *g* for 30 min at 4 °C, and the total protein concentration was measured in the supernatant (Bradford colorimetric method; Merck, Darmstadt, Germany). 

### 2.8. BV2 Cells

A murine microglial line BV2 (mouse, C57BL/6, brain, microglial cells, Tema Ricerca, Genova, Italy) was used for this study. The cells were thawed and placed in a 75 cm^2^ flask (Sarstedt, Milan, Italy) in a medium containing RPMI with the addition of 10% of heat-inactivated (56 °C, 30 min) fetal bovine serum (FBS, Gibco^®^, Milan, Italy) and 1% glutamine. Cells were grown at 37 °C and 5% CO_2_ with daily medium change [39].

#### 2.8.1. Cells Treatments and Neuroinflammation Model

Cells were then pretreated with COR167 and suberoylanilide hydroxamic acid (SAHA, 5 µM) for 4 h and then stimulated with Lipopolysaccharide (LPS, Sigma-Aldrich, Italy) 250 ng mL^−1^ for 24 h. Both substances were dissolved in DMSO 1% in saline.

#### 2.8.2. Sulforhodamine B (SRB) Assay

The SRB test was used to assess the cell viability. Briefly, cells were seeded in 96-well plates (2 × 10^4^ cells per well). After treatment, cells were fixed by adding 50% trichloroacetic acid (Merck, Darmstadt, Germany) in RPMI to the wells and the plate was incubated at 4 °C for 1 h. Then, the plate was gently washed with water and allowed to dry for 1 h before staining with 30 µL of SRB stain (4 mg/mL solution) in 1% acetic acid in double distilled H_2_O at rt. After 30 min, the plate was washed 4/5 times with acetic acid to remove excess stain. A volume of 200 µL of Tris-HCl buffer (pH = 10) was added, and the plate was placed on a shaker for 5 min. The absorbance was determined spectrophotometrically at 570 nm using a multiplate reader (Biorad, Milan, Italy). Three independent experiments (*n* = 3) were carried out for each treatment. Cell viability values were normalized to the mean of the control.

#### 2.8.3. BV2 Cell Lysate

Microglial cells were seeded in 6-well plates (3 × 10^5^ cells/well) until 70–80% confluence was achieved. Protein extraction from tissues and cells was performed as described [42]. Briefly, proteins from BV2 cells were extracted by radioimmunoprecipitation assay (RIPA) buffer (50 mM Tris-HCl pH 7.4, 150 mM NaCl 1% sodium deoxycholate, 1% Tryton X-100, 2 mM PMSF) (Sigma-Aldrich, Italy). After homogenization, the samples were spun at 12,000× *g* for 30 min, 4 °C, the supernatants were collected, and the insoluble pellet was separated. The total protein concentration was measured in a portion of each supernatant using Bradford colorimetric method (Sigma-Aldrich, Milan, Italy).

### 2.9. Immunofluorescence

Mice were perfused transcardially with 4% paraformaldehyde in 0.1 M phosphate-buffered saline (PBS, pH 7.4) on day 7. Thereafter, the lumbar spinal cord was quickly removed, postfixed for 18 h with the same fixative at 4 °C, and transferred to 10%, then 20%, and then 30% sucrose solution. After preincubation in 5 mg/mL bovine serum albumin (BSA)/0.3% Triton-X-100/PBS, sections were incubated overnight at 4 °C with the primary antibodies as follows: HDAC1 (1:100; Santa Cruz Biotechnology, Dallas, TX, USA) and IBA-1 (1:100; Santa Cruz Biotechnology, Dallas, TX, USA). After rinsing in PBS containing 0.01% Triton-X-100, sections were incubated in secondary antibodies labeled with Invitrogen Alexa Fluor 488 (1:400; Thermo Fisher Scientific), Invitrogen Alexa Fluor 568 (1:400; Thermo Fisher Scientific) at room temperature for 2 h. Sections were coverslipped using Vectorshield mounting medium (Vector Laboratories, Burlingame, CA, USA). Representative images were acquired through a Leica DM6000B fluorescence microscope equipped with a DFC350FX digital camera with appropriate excitation and emission filters for each fluorophore. Images were acquired with 10× to 40× objectives using a digital camera. The immunofluorescence intensity was calculated by ImageJ (Wayne Rasband, National Institute of Health, USA) [43]. 

### 2.10. Western Blotting

Protein samples (40 µg of protein/lane) were separated by SDS-PAGE on 10% minigel [44]. Thereafter, proteins were then transferred to nitrocellulose membranes for 120 min at 100 V. After blocking (120 min in PBST containing 5% non-fat dry milk), membranes were incubated overnight at 4 °C with primary antibodies: HDAC1 (1:1000), anti-IKBα (1:1000), and anti-IL10R (1:1000) (Santa Cruz Biotechnology). Blots were then rinsed three times with PBST and incubated at room temperature for 2 h with HRP-conjugated mouse anti-rabbit (1:3000) and goat anti-mouse (1:5000, Bioss Antibodies, Woburn, MA, USA). A chemiluminescence detection system (Pierce, Milan, Italy) was used, and signal intensity (pixels/mm^2^) was quantified using ImageJ (NIH). GAPDH (1:5000, sc-32233) (Santa Cruz Biotechnology, Dallas, TX, USA) was used to normalize the signal intensity.

### 2.11. Statistical Analysis

Results are reported as mean ± SEM. Behavioral tests: a one-way analysis of variance (ANOVA) followed by the Tukey post hoc test and a two-way ANOVA followed by the Bonferroni post hoc test were used for statistical analysis. Western blotting experiments: 5 mice per treatment group were included, and each run was in triplicate. The differences between groups were determined by one-way ANOVA followed by the Tukey post hoc test. In immunofluorescence experiments, immunoreactive areas are mean values of 5 separate experiments, and differences among mean immunoreactive areas were analyzed by one-way ANOVA, followed by the Tukey post hoc test. *p* value less than 0.05 was considered significant, and analyses were performed through GraphPad Prism version 9.5 (GraphPad Software Inc., San Diego, CA, USA).

## 3. Results

### 3.1. Analgesic Effect of COR167 Registered in the Hot Plate Test

The analgesic effect of COR167 in acute pain conditions was tested on the hot plate. COR167 was administered orally at 3, 10, 30, and 100 mg/kg, and the time course of each dose was recorded at baseline (BL) and then at 30, 60, 90, 120, 150, and 180 min after administration (Figure 1A). COR167 3 mg/kg did not induce an increase of algic threshold to any time registered compared to the baseline. COR167 10 mg/kg after oral administration increased the latency to heat response after 60 min from the administration. COR167 30 mg/kg showed its peak activity at 90 min, which disappeared at 120 min. COR167 100 mg/kg has a peak activity shifted to 120 min post-administration (Figure 1B).

### 3.2. Mechanical Allodynia in SNI Mice Was Reduced by COR167 after a Single Oral Administration

The antinociceptive activity of COR167 was investigated in a condition of neuropathic pain. COR167 3–100 mg/kg was tested after 7 days post-induction of spared nerve injury (SNI) model (Figure 2A), which is a model of peripheral mononeuropathy that induced strong mechanical allodynia starting from 3 days post-surgery [39]. To measure the mechanical allodynia, we used the von Frey filaments 0.07–2.00 g. Coherently with the hot plate test, oral administration of 3 mg/kg (Figure 2B) did not alter the allodynia produced by the SNI model in the injured *ipsi* hind paw, compared to the uninjured *contra* hind paw. COR167 10 mg/kg (Figure 2C) reduced the gap of mechanical allodynia between *contra* and *ipsi* after 60 min from oral administration, but the effect disappeared immediately after 90 min. COR167 30 mg/kg (Figure 2D) showed a similar trend to COR167 10 mg/kg, with a peak of the effect after 60 min from the administration. Finally, the dose of 100 mg/kg (Figure 2E) showed the same tendency but with an efficacy also on the *contra* hind paw after 60 min from administration. Then, to normalize the final effect against allodynia, we calculated the “average absolute threshold”, which is the difference between *contra* and *ipsi* hind paw values registered at different times. As reported in Figure 2F, we observed that only the oral administration of COR167 10 mg/kg drastically reduced the differences between *contra* and *ipsi* values, generating an anti-hypersensitivity effect (60 min after administration; 3 mg/kg: 0.70 ± 0.26; 10 mg/kg: 0.24 ± 0.05; 30 mg/kg: 0.60 ± 0.13; 100 mg/kg: 0.56 ± 0.56). 

### 3.3. Repeated Oral Administration of COR167 10 mg/kg Reduced Symptoms Associated with the SNI Model after 14 Days Post-Surgery

Results obtained following single-dose administration encouraged us to continue investigating the activity of the 10 mg/kg dose following repeated administration. Mice were treated from post-operative day 3 to 14 (Figure 3A). On the 14th day, we measured the mechanical threshold in both hind paws (Figure 3B) (*contra* and *ipsi*), the response to thermal hyperalgesia (Figure 3C), motor coordination (Figure 3D), spontaneous locomotor activity (Figure 3E), and anxiety behavior (Figure 3F). The control group (VEH) showed persistent mechanical allodynia in the *ipsi*-lateral side compared to the *contra* on post-surgery day 14, as previously observed [39]. Repeated oral administration of COR167 significantly reduced the allodynia in the ipsilateral side (Figure 3A), with an efficacy comparable to that produced by pregabalin (PREG), a widely employed treatment in the management of neuropathic pain, used as a reference drug. Concerning thermal hyperalgesia, the VEH group showed lower values of latency to licking in the ipsilateral side compared to the contralateral side (13 ± 0.5), which remains stable for the time course. COR167 showed a tendency to reduce thermal hyperalgesia after 30 min, which became significant after 60 min and disappeared after 90 min from administration (Figure 3C). The effect observed at 60 min was of intensity comparable to that shown by MORPH after 30 min of oral administration, which represents the peak of this analgesic drug (Figure 3C). Repeated oral administration of COR167 did not produce an alteration of motor coordination in the rotarod test; indeed, there was not a significant difference in the number of falls compared to the control group (Figure 3D). In the hole board test, COR167 did not show variations in the number of intrusions in the holes or movements on the plane, excluding possible side effects on spontaneous mobility (planes) and exploratory activity (holes) (Figure 3E). Moreover, COR167 increased the time spent in the center of the box in the open field test compared to the untreated group showing an anxiolytic-like activity (Figure 3F). 

### 3.4. COR167 Attenuated Neuroinflammation via HDAC-1 Reduction in the SNI Dorsal Horn Spinal Cord

SNI is characterized by spinal neuroinflammation with a selective increase in HDAC-1 protein expression in the ipsilateral side of SNI mice spinal cord [32,45]. Consistently with previous observations, Figure 4A shows the colocalization of HDAC-1 (green) with IBA-1 (red), a widely used marker of microglia cells, and both of them are overexpressed in the ipsilateral side of SNI mice compared to the contralateral side. Immunofluorescence images and quantification analysis showed the reduction of HDAC-1 protein expression in the *ipsi* dorsal horn of SNI-treated mice, compared to the VEH group, after administration of COR167 10 mg/kg (Figure 4B). Western blot experiments confirmed the HDAC1 overexpression in SNI spinal cord preparations and the prevention of this effect by COR167 treatment with an intensity similar to that produced by SAHA, a well-known HDAC inhibitor (Figure 4C). COR167 10 mg/kg reduced the mechanical allodynia in SNI mice with the same time course observed for SAHA. Both compounds showed a peak of the effect at 60 min that persisted after 90 min and completely disappeared at 120 min after administration (Figure 4D). HDAC-1 is involved in the activation of microglia in the pro-inflammatory state, leading to an up-regulation of the NF-kBp65 activation pathway [33,46] and a reduction of anti-inflammatory cytokines, such as IL-10 [47]. SNI mice showed a reduction of IkBα (Figure 4D), an inhibitory protein of NF-kBp65 cytosolic fraction, and IL-10 (Figure 4E) in the ipsilateral side of the spinal cord tissue compared to the contralateral side. COR167 prevented the activation of the NF-kBp65 pathway, counteracting the IkBα (Figure 4D) and IL-10 (Figure 4E) reduction, leading them to values similar to the contralateral side.

### 3.5. COR167 Prevented Microglia Activation and Reduced HDAC1 Expression in BV2 Cells

To confirm the effect of COR167 on HDAC1 expression, we used a standardized in vitro neuroinflammation model using murine microglia BV2 cells stimulated with LPS 250 ng/mL for 24 h. As previously reported [46], we observed that BV2 selectively expressed HDAC1 more than other isoforms after 24 h of LPS stimulation. COR167 was tested at different concentrations to establish the maximum non-toxic concentration of 10 μM (Figure 5A). BV2 are morphologically dynamic cells; indeed, the changing of their shape influenced their biological activity. LPS induced an ameboid-like shape with short ramification, highlighting a “reactive” state (Figure 5B). COR167-pretreated BV2 assumed a morphology similar to that observed in the CTRL group (Figure 5B). LPS induced a reduction of cell viability compared to the untreated cells, which is significantly attenuated by COR167. The co-pretreatment of COR167 with the CB2 antagonist AM630 (1 µM) completely prevented the cytoprotective effect of COR167 (Figure 5C). The increase of HDAC1 expression produced by LPS was prevented by COR167, confirming the data obtained in the spinal cord of SNI animals, with an efficacy comparable to that produced by SAHA 5 µM. This effect on HDAC1 expression was prevented by AM630, demonstrating a CB2-dependent effect.

## 4. Discussion

The selective stimulation of CB2 receptors is increasingly recognized as a safer novel therapeutic approach for the treatment of neuropathic pain conditions due to the lack of centrally mediated unwanted effects associated with the activation of CB1 receptors. In addition to the positive effects induced by CB2 acting natural constituents [25], several CB2 agonist compounds showed anti-neuropathic activity. Intrathecal administration of JWH-015 attenuated nerve injury-induced allodynia in the lumbar five nerve transection neuropathic pain model in rats [47]. Self-administration of JWH-133 attenuated spontaneous pain in the partial sciatic nerve ligation model of neuropathic pain in the absence of reinforcing effects in animals without pain [48]. GW405833 [49] and AM1241 [50] attenuated mechanical and thermal hypersensitivity in the chronic constriction injury model in rats. In the efforts to find a new effective and safe treatment for neuropathic pain, this present study investigated the antinociceptive, anti-hyperalgesic, and anti-allodynic properties of COR167 after acute or repeated treatment in mice. 

A single oral administration of COR167 induced thermal antinociception in naïve mice. These findings confirm and extend previous studies showing the analgesic effect of COR167 in the formalin test of acute peripheral and inflammatory pain [29] and are consistent with the literature evidence on the pain-relieving activity of selective CB2 agonist compounds. Indeed, intraperitoneal and intraplantar administration of AM1241 induced thermal antinociception in the plantar test in rats [51] and in the tail flick and hot plate tests in mice [52]. In this present study, we additionally present evidence on the attenuation of mechanical allodynia and thermal hyperalgesia on trauma-induced neuropathic pain after an acute oral administration. Comparable results were also obtained after COR167 repeated treatment. This reversal of neuropathic pain-associated allodynia and hyperalgesia without tolerance development appears peculiar for CB2-mediated anti-neuropathic activity since it has been demonstrated for several selective CB2 agonists such as GW405833 [53] and JW015 [12], in contrast to treatment with mixed CB1/CB2 agonist. Along with pain-relieving effects, COR167 repeated administration showed an anxiolytic-like activity indicating the capability to attenuate comorbidities associated with neuropathic pain states.

Evidence indicates that reactive microglia express CB2 receptors [10,11,12]. An increase in the protein and mRNA levels of spinal CB2 receptors occurs in neuropathic pain conditions [54,55], indicating the involvement of these receptors in neuropathy-associated spinal neuroinflammation. Microglia-mediated neuroinflammation relies on transcription factors such as NF-κB, widely considered the driver of microglia. NF-κB regulates inflammatory gene transcription, and it is regulated by lysine acetylation [56]. Studies indicate that some class I HDAC members could have a prominent role. It has been demonstrated that HDAC1 protein levels are increased in the spinal cord of neuropathic mice, and the administration of selective HDAC1 inhibitors attenuated neuropathic pain symptomatology [31,57] and reduced the activation of NF-κB [32,45]. Consistently, we found an overactivation of spinal microglia in the ipsilateral side of SNI mice, along with an overexpression of HDAC1 in microglial cells. Treatment of SNI mice with COR167 at anti-neuropathic doses prevented HDAC1 overexpression and NF-κB activation, along with an increase in the anti-inflammatory cytokine IL-10, through a CB2-mediated mechanism, as confirmed by experiments conducted in BV2 cells.

## 5. Conclusions

Due to the key role of spinal microglia in the regulation of central sensitization, targeting spinal microglial CB2 as a mechanism to produce control over the aberrant neuroinflammatory response can serve as a promising therapeutic approach for pain relief. Dampening neuroinflammation by modulating CB2 activity may result in antinociceptive and neuroprotective effects [58], making CB2 agonist of therapeutic value in conditions of neuropathic pain. We demonstrated that the oral administration of the selective CB2 agonist COR167 reduced neuropathic pain symptoms induced by SNI in mice. These effects appeared related to an HDAC1-mediated attenuation of spinal neuroinflammation that leads to a reduction in the NF-kB activation. Oral administration of COR167 might represent an innovative therapeutic perspective in the management of neuropathic pain conditions.

## Figures and Tables

**Figure 1 biomedicines-11-01546-f001:**
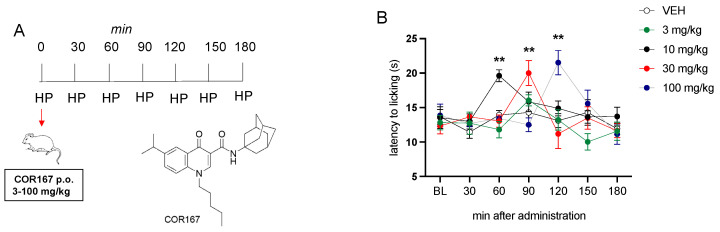
Analgesic activity of COR167 in acute thermal pain. (**A**) Administration and test schedule. (**B**) Dose-response curve for the analgesic activity of COR167 (3–100 mg/kg p.o.) in the hot plate test. ** *p* < 0.01 versus the before-treatment licking latency values (BL).

**Figure 2 biomedicines-11-01546-f002:**
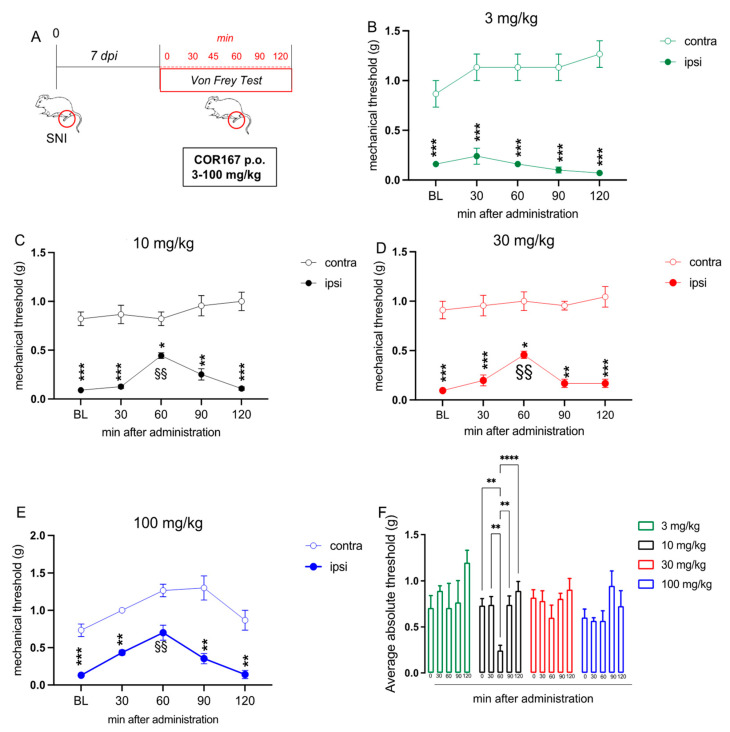
COR167 attenuation of neuropathic pain after acute administration. (**A**) Experimental protocol. Tests were performed on day 7 post-surgery. (**B**) COR167 3 mg/kg p.o. was ineffective. (**C**–**E**) Dose-dependent attenuation of mechanical allodynia by COR167 (10–30–100 mg/kg p.o. (**F**) COR167 10 mg/kg p.o. showed the most prominent antiallodynic activity through the evaluation of the average absolute threshold. * *p* < 0.05, ** *p* < 0.01, *** *p* < 0.001, **** *p* < 0.0001 vs. contralateral side. §§ *p* < 0.01 vs. BL.

**Figure 3 biomedicines-11-01546-f003:**
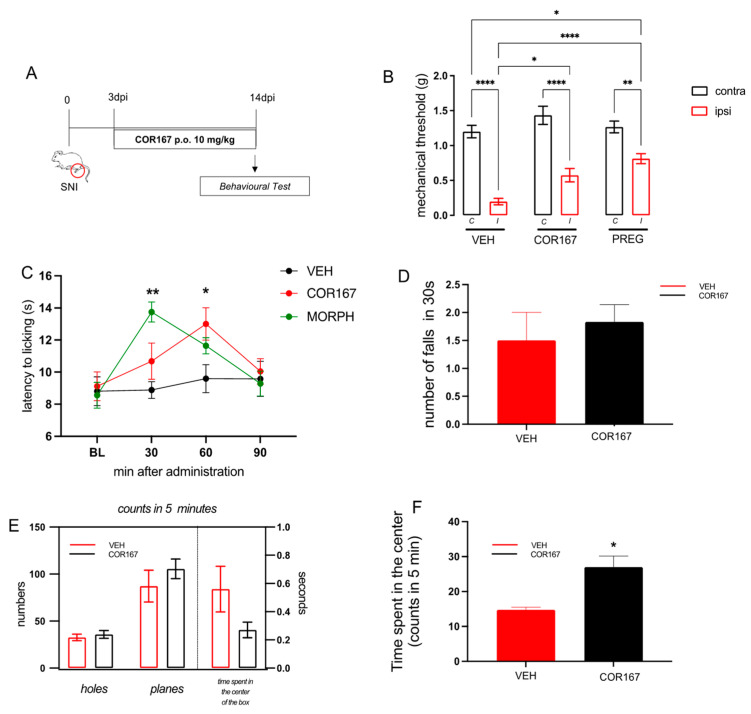
COR167 attenuation of neuropathic pain after repeated administration. (**A**) Experimental protocol. COR167 10 mg/kg p.o. was administered daily from day 3 to 14 post-injury. Tests were performed on day 14 post-injury. (**B**) COR167 attenuation of mechanical allodynia in the ipsilateral side (ipsi) with an efficacy comparable to the reference drug pregabalin (PREG). * *p* < 0.05, ** *p* < 0.01, **** *p* < 0.0001 vs. contralateral side (contra). (**C**) Time-course curve for the attenuation of thermal hyperalgesia by COR167 compared to morphine (MORPH). * *p* < 0.05, ** *p* < 0.01 vs. vehicle-treated mice. (**D**) Lack of impairment of motor coordination by COR167 repeated treatment in the rotarod test. (**E**) COR167 did not alter spontaneous mobility (planes) and exploratory activity (holes) in the hole board test. (**F**) Increase of the time spent in the center of the arena by COR167 in the open field test. * *p* < 0.05 vs. vehicle-treated mice.

**Figure 4 biomedicines-11-01546-f004:**
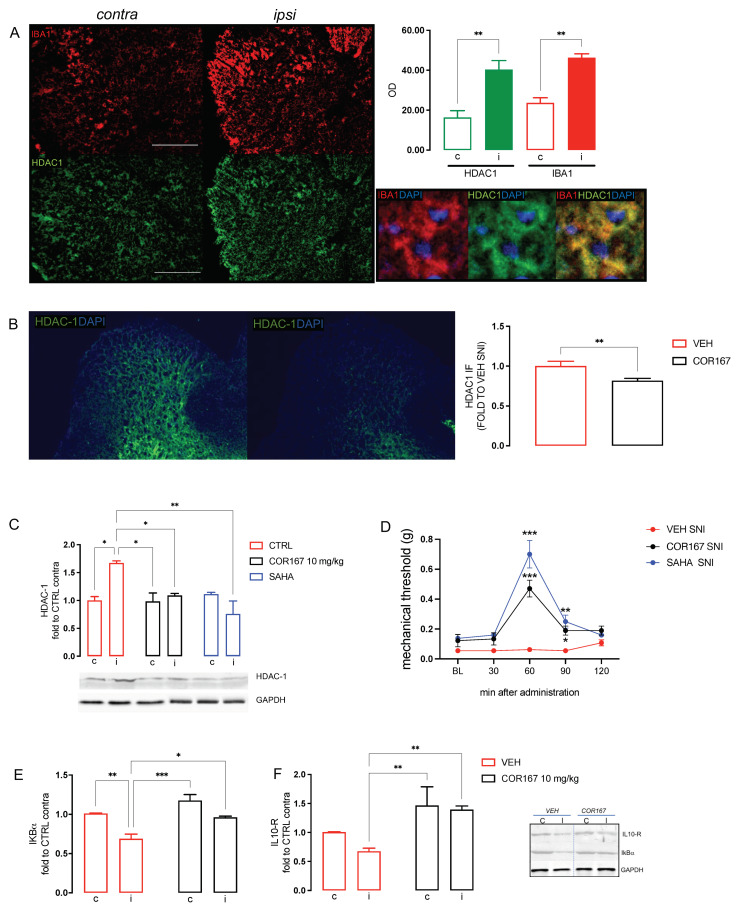
COR167 attenuation of spinal neuroinflammation via an HDAC1-mediated mechanism. (**A**) Over-expression of IBA1 (red) and HDAC1 (green) in the ipsilateral side (I) of SNI mice spinal cord compared to the contralateral side (**C**) and their co-localization. ** *p* < 0.01 vs. contralateral side. (**B**) COR167 10 mg/kg p.o. repeated administration reduced HDAC1 overexpression in the ipsilateral side (ipsi) of SNI mice in immunofluorescence experiments. ** *p* < 0.01 vs. vehicle. (**C**) HDAC1 overexpression in the ipsilateral side was reduced by COR167 treatment with an efficacy comparable to SAHA (10 mg/kg i.p.). * *p* < 0.05, ** *p* < 0.01. (**D**) Time-course curves for COR167 and SAHA in the von Frey test. * *p* < 0.05, ** *p* < 0.01, *** *p* < 0.001 vs. vehicle. (**E**) Prevention of IkBα decrease by COR167. * *p* < 0.05, ** *p* < 0.01, *** *p* < 0.001. (**F**) Increase of the IL-10 protein expression by COR167 treatment. ** *p* < 0.01.

**Figure 5 biomedicines-11-01546-f005:**
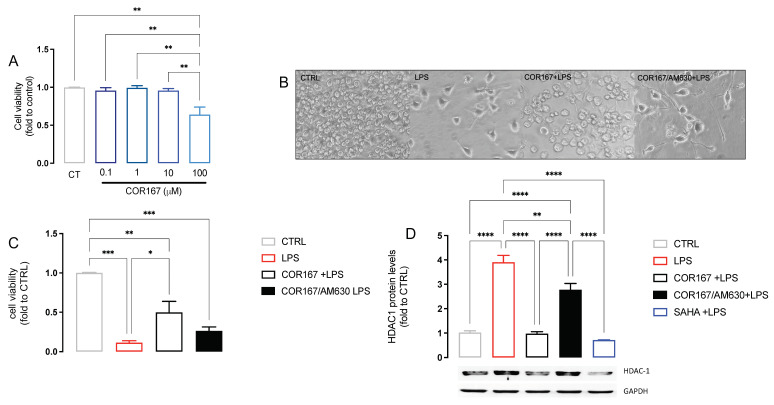
Attenuation of neuroinflammation by COR167 in LPS-stimulated BV2 cells. (**A**) COR167 dose-response curve (0.1–100 µM) for cell viability in BV2 cells. CT: control untreated cells. ** *p* < 0.01. (**B**) Ameboid-like morphology of LPS-stimulated BV2 cells that was prevented by COR167 10 µM pretreatment. AM630 (1 µM) prevented the COR167 effect. (**C**) COR167 attenuated the reduction of cell viability induced by LPS exposure. This effect was antagonized by AM630 co-treatment. * *p* < 0.05, ** *p* < 0.01, *** *p* < 0.001. (**D**) COR167 abolished the LPS-induced HDAC1 overexpression in BV2 cells with an intensity comparable to SAHA 5µM. AM630 co-treatment completely antagonized this effect. ** *p* < 0.01, **** *p* < 0.0001.

## Data Availability

The data presented in this study are available on request from the corresponding author.

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
