# Peer review of "The Selective CB2 Agonist COR167 Reduced Symptoms in a Mice Model of Trauma-Induced Peripheral Neuropathy through HDAC-1 Inhibition"

_biomedicines, 2023, doi:10.3390/biomedicines11061546_

Round 1
Reviewer 1 Report
Manuscript (biomedicines-2373894) by Borgonetti et al found that oral administration of a selective CB2 agonist COR167 inhibited neuropathic pain effectively and reduced anxiety in several mouse behavioral test paradigms of pains, locomotor, and anxiety. They demonstrated that activation of microglia on SNI ipsilateral dorsal horn spinal cord was reduced by oral COR167 through suppression of HDAC1 overexpression and reduction of neuroinflammation markers. They compared COR167 effect with the reference analgesics of pregabalin and morphine and HDAC1 inhibitor SAHA to support they conclusion. The dose responses and time course of the treatments were done scientifically and microglia cell culture BV2 supported their in vivo study. My main concern is whether other cell types of the spinal cord are also involved in the suppression of HDAC1 and neuroinflammation.
1. SNI induced HDAC1 and the suppression of HDAC1 by COR167 in dorsal horn were not restricted to microglia as shown in Fig 4A and 4B.
2. Since all the CB2 antibodies are not specific for IHC, I suggest the authors should use RNAscope probes for multiplex in situ hybridization in the ipsi and contra sites of spinal cord of SNI model to find out whether neuron and astrocytes are also involved. Here are catalog probes information from ACDbio. CB2R: RNAscope™ Probe- Mm-Cnr2 (Cat No. 407351), Microglia: RNAscope™ Probe- Mm-Aif1-C2 (Cat No. 319141-C2), Neuron: RNAscope™ Probe- Mm-Rbfox3-C2 (Cat No. 313311-C2), Astrocyte: RNAscope™ Probe- Mm-Gfap-O1-C3 (Cat No. 481691-C3).
3. Explain why oral administration of COR167 is more effective than JWH133?
Author Response
Manuscript (biomedicines-2373894) by Borgonetti et al found that oral administration of a selective CB2 agonist COR167 inhibited neuropathic pain effectively and reduced anxiety in several mouse behavioral test paradigms of pains, locomotor, and anxiety. They demonstrated that activation of microglia on SNI ipsilateral dorsal horn spinal cord was reduced by oral COR167 through suppression of HDAC1 overexpression and reduction of neuroinflammation markers. They compared COR167 effect with the reference analgesics of pregabalin and morphine and HDAC1 inhibitor SAHA to support they conclusion. The dose responses and time course of the treatments were done scientifically and microglia cell culture BV2 supported their in vivo study. My main concern is whether other cell types of the spinal cord are also involved in the suppression of HDAC1 and neuroinflammation.
We thank the reviewer for her/his comments, and we hope that with our answers can solve her/his concerns.
- SNI induced HDAC1 and the suppression of HDAC1 by COR167 in dorsal horn were not restricted to microglia as shown in Fig 4A and 4B.
Reply: in addition to microglia, both HDAC1 and CB2 receptors are expressed in other cell types, and we cannot exclude that COR167 might influence HDAC1 expression in cells other than microglia. We have reported in previously works that the SNI model increased HDAC-1 expression in the dorsal horn of spinal cord tissue (DOI: 10.1016/j.ejphar.2018.02.034; doi: 10.1016/j.pbb.2017.08.006.), particularly in microglia cell (doi: 10.1016/j.phrs.2021.105431, doi: 10.1016/j.phymed.2023.154670, doi: 10.1016/j.phymed.2020.153307). Indeed, we demonstrated that the over expression of HDAC 1 lead spinal microglia and BV-2 (murine microglia cell line) to a proinflammatory state, with an increase of NF-kBp65 nuclear translocation and up-regulation of inflammatory cytokines. In addition, CB2 microglial receptors are abundantly overexpressed in inflammatory states and pathological conditions. Thus, our hypothesis is that microglia have a prominent role and might represent a main target for COR167. The complete abolishment of LPS-induced HDAC1 overexpression in BV2 cells, as reported in Figure 5D, further support our hypothesis.
- Since all the CB2 antibodies are not specific for IHC, I suggest the authors should use RNAscope probes for multiplex in situ hybridization in the ipsi and contra sites of spinal cord of SNI model to find out whether neuron and astrocytes are also involved. Here are catalog probes information from ACDbio. CB2R: RNAscope™ Probe- Mm-Cnr2 (Cat No. 407351), Microglia: RNAscope™ Probe- Mm-Aif1-C2 (Cat No. 319141-C2), Neuron: RNAscope™ Probe- Mm-Rbfox3-C2 (Cat No. 313311-C2), Astrocyte: RNAscope™ Probe- Mm-Gfap-O1-C3 (Cat No. 481691-C3).
Reply: we thank the reviewer for the suggestions and information.
- Explain why oral administration of COR167 is more effective than JWH133?
Reply: in the present study we did not compare the efficacy of COR167 to that of JWH133. Furthermore, to the best of our knowledge, there is no literature data on the efficacy of JWH133 in the SNI model to compare with our findings to evaluate the relative efficacy.

Reviewer 2 Report
Type of manuscript: Article
Title: The selective CB2 agonist COR167 reduced symptoms in a mice model of trauma-induced peripheral neuropathy through HDAC-1 inhibition
Journal: Biomedicines
The work is scientifically accurate, well developed and fluently described. Substantially, it fit the scope of the journal and provides new insight regarding the selective CB2 agonist COR167 in a mice model of trauma-induced peripheral neuropathy through HDAC-1 inhibition.
Minor revisions:
Page 1, Abstrac, line 16:
it could be better to change “neuropathic patients” with “neuropathic pain”.
Page 2, line 79:
“Pregabalin (PREG 30 mg/kg i.p; Sigma Aldrich), used as a reference drug for mechanical allodynia, was dissolved in saline and administered orally 3 h before testing”.
Pregbalin is given i.p. or orally?
Results page 5, line 254:
“COR167 10 mg/kg after oral administration increased the latency to heat response after 60 min from the administration. COR167 255 30 mg/kg showed its peak activity at 90 min, which disappeared at 120 min. COR167 100 mg/kg has a peak activity shifted to 120 minutes post-administration”.
How do the authors explain the difference in analgesia peaks? 10mg administration gives a peak at 60 min, 30mg administration at 90min and 100mg administration at 120min.
Figure 1(B) Dose-response curve for the analgesic activity of COR167 (3-100 mg/kg p.o.) in the hot plate test.
It would have been appropriate for the authors to insert the trend of the thresholds recorded after the administration of the vehicle
Page 6, line 270:
“…is a model of peripheral mononeuropathy that induced strong mechanical and thermal hyperalgesia starting from 3 days post-surgery…”
The authors report the data regarding hyperalgesia and then comment instead on allodynia. It would be better to report a data relating to allodynia.
Page 6, line 276:
the citation of figure 2D is missing from the text.
Page 6 line 277: in this test, the antiallodinic peak is always at 60min.
Could the authors comment on this data?
Page 7 line 302
“Repeated oral administration of COR167 significantly reduced the allodynia in the ipsilateral side (Figure 3A), with an efficacy comparable to that produced by pregabalin, a widely employed treatment in the management of neuropathic pain, used reference drug”.
Do the authors want to demonstrate that repeated administration of the compound prevents the manifestation of allodynia?
Author Response
The work is scientifically accurate, well developed and fluently described. Substantially, it fit the scope of the journal and provides new insight regarding the selective CB2 agonist COR167 in a mice model of trauma-induced peripheral neuropathy through HDAC-1 inhibition.
We thank the reviewer for her/his positive comment and for improving the manuscript with her/his suggestion.
Minor revisions:
Page 1, Abstrac, line 16: it could be better to change “neuropathic patients” with “neuropathic pain”.
Reply: We changed it.
Page 2, line 79:
“Pregabalin (PREG 30 mg/kg i.p; Sigma Aldrich), used as a reference drug for mechanical allodynia, was dissolved in saline and administered orally 3 h before testing”. Pregbalin is given i.p. or orally?
Reply: Pregabalin was administered i.p., we thank the reviewer to bring this typo to our attention.
Results page 5, line 254:
“COR167 10 mg/kg after oral administration increased the latency to heat response after 60 min from the administration. COR167 255 30 mg/kg showed its peak activity at 90 min, which disappeared at 120 min. COR167 100 mg/kg has a peak activity shifted to 120 minutes post-administration”.
How do the authors explain the difference in analgesia peaks? 10mg administration gives a peak at 60 min, 30mg administration at 90min and 100mg administration at 120min.
Reply: we don’t have a definitive answer to this question. Since there was no improvement in the efficacy from 30 to 100 mg, we hypothesized the involvement of a pharmacokinetic mechanism that is currently under investigation.
Figure 1(B) Dose-response curve for the analgesic activity of COR167 (3-100 mg/kg p.o.) in the hot plate test.
It would have been appropriate for the authors to insert the trend of the thresholds recorded after the administration of the vehicle
Reply: According to reviewer suggestion we added to the graph in Figure 1B the vehicle group.
Page 6, line 270:
“…is a model of peripheral mononeuropathy that induced strong mechanical and thermal hyperalgesia starting from 3 days post-surgery…”
The authors report the data regarding hyperalgesia and then comment instead on allodynia. It would be better to report a data relating to allodynia.
Reply: The Von frey filaments are the most widely used test for measuring allodynia in neuropathic pain both in clinical (doi: 10.1016/j.sjpain.2010.08.001, doi: 10.1017/S0265021507000221, doi: 10.1097/01.ajp.0000210950.01503.72) and preclinical model (doi: 10.3389/fnmol.2017.00284, doi: 10.1097/PR9.0000000000000824, doi: 10.1089/can.2022.0096). To make it clearer, we replaced the term hyperalgesia with allodynia.
Page 6, line 276: the citation of figure 2D is missing from the text.
Reply: We thank the review to bring this typo on our attention.
Page 6 line 277: in this test, the antiallodinic peak is always at 60min. Could the authors comment on this data?
Reply: observing the effect produced by COR167 at 100 mg, condition in which an increase of the pain threshold was produced also in the contralateral side, there was a peak of the antiallodynic activity at 60 min whereas in the contra the peak occurred at 90 min. Since contra and ipsi values were recorded in the same mouse, this different time-course let hypothesize a different mechanism involved in the promotion of an antiallodynic (ipsi) or analgesic (contra) effect.
Page 7 line 302
“Repeated oral administration of COR167 significantly reduced the allodynia in the ipsilateral side (Figure 3A), with an efficacy comparable to that produced by pregabalin, a widely employed treatment in the management of neuropathic pain, used reference drug”.
Do the authors want to demonstrate that repeated administration of the compound prevents the manifestation of allodynia?
Reply: The principal aim of this work is to demonstrate that the oral administration of a selective CB2 agonist can reduce allodynia through its anti-neuroinflammatory activity at the spinal level, and prevent further damage produced by the neuroinflammatory process induced by the SNI model. We performed both acute and repeated administration to evaluate a potential reduction/loss of antiallodynic efficacy after repeated treatment (comparison with pregabalin and morphine) as well as the occurrence of side effects (locomotor behaviour) not visible after a single administration.

Round 2
Reviewer 1 Report
Future experiments to compare efficacy of COR167 to JWH133oral administration on DNI model. Other cell types of spinal cord should be investigated.